# Non-Invasive Wearable Devices for Monitoring Vital Signs in Patients with Type 2 Diabetes Mellitus: A Systematic Review

**DOI:** 10.3390/bioengineering10111321

**Published:** 2023-11-16

**Authors:** Artur Piet, Lennart Jablonski, Jennifer I. Daniel Onwuchekwa, Steffen Unkel, Christian Weber, Marcin Grzegorzek, Jan P. Ehlers, Olaf Gaus, Thomas Neumann

**Affiliations:** 1Institute of Medical Informatics, University of Lübeck, 23562 Lübeck, Germany; 2Department of Digital Health Sciences and Biomedicine, University of Siegen, 57076 Siegen, Germany; 3Department of Medical Statistics, University Medical Center Göttingen, 37075 Göttingen, Germany; 4Department of Knowledge Engineering, University of Economics in Katowice, 40-287 Katowice, Poland; 5Department of Didactics and Educational Research in Health Science, Witten/Herdecke University, 58455 Witten, Germany; 6Faculty of Economics and Management, Otto von Guericke University Magdeburg, 39106 Magdeburg, Germany; 7University Department of Neurology, Otto von Guericke University Magdeburg, 39106 Magdeburg, Germany

**Keywords:** type 2 diabetes, RCTs, vital signs, sensing technology

## Abstract

Type 2 diabetes mellitus (T2D) poses a significant global health challenge and demands effective self-management strategies, including continuous blood glucose monitoring (CGM) and lifestyle adaptations. While CGM offers real-time glucose level assessment, the quest for minimizing trauma and enhancing convenience has spurred the need to explore non-invasive alternatives for monitoring vital signs in patients with T2D. **Objective:** This systematic review is the first that explores the current literature and critically evaluates the use and reporting of non-invasive wearable devices for monitoring vital signs in patients with T2D. **Methods:** Employing the PRISMA and PICOS guidelines, we conducted a comprehensive search to incorporate evidence from relevant studies, focusing on randomized controlled trials (RCTs), systematic reviews, and meta-analyses published since 2017. Of the 437 publications identified, seven were selected based on predetermined criteria. **Results:** The seven studies included in this review used various sensing technologies, such as heart rate monitors, accelerometers, and other wearable devices. Primary health outcomes included blood pressure measurements, heart rate, body fat percentage, and cardiorespiratory endurance. Non-invasive wearable devices demonstrated potential for aiding T2D management, albeit with variations in efficacy across studies. **Conclusions:** Based on the low number of studies with higher evidence levels (i.e., RCTs) that we were able to find and the significant differences in design between these studies, we conclude that further evidence is required to validate the application, efficacy, and real-world impact of these wearable devices. Emphasizing transparency in bias reporting and conducting in-depth research is crucial for fully understanding the implications and benefits of wearable devices in T2D management.

## 1. Introduction/Background

Telehealth applications, such as remote monitoring, hold considerable promise in the management of chronic diseases such as type 2 diabetes (T2D) [1]. Telehealth applications are particularly useful when an effective treatment of the disease entails extensive changes in the patient’s lifestyle. By helping patients self-manage their condition daily, these applications can effectively reduce complications, improve the course of the disease, and improve the quality of life of patients [2]. The cause of T2D is primarily due to an unhealthy diet and insufficient physical activity, which can lead to excess body weight, obesity, and the ineffective use of insulin by the body [3]. This disease can cause severe complications, including damage to small and large blood vessels and nerves, leading to loss of vision, kidney failure, heart attacks, strokes, and lower limb amputations [4].

In 2021, the estimated prevalence of diabetes (both type 1 and type 2) in the age group of 20 to 79 was 10.7% in the US [5] and 9.1% in Europe [6], making it a global health challenge. In the same year, diabetes-related expenditures totaled USD 189.3 billion, and in 2019, approximately 2 million deaths globally resulted from T2D [6]. The World Health Organization (WHO) recommends lifestyle changes, including a healthy diet, regular physical activity, maintaining proper body weight, avoiding tobacco, and minimizing alcohol consumption as key preventive measures for T2D. Based on evidence from various studies in a recent review, the authors [7] showed that almost normalizing glycemic control can be achieved if patients lose around 15% of their body weight, and such a weight reduction was seen in more than 25% of patients with T2D. As a result, effective self-management of T2D incorporates these lifestyle changes as well as CGM.

Although continuous glucose monitoring (CGM) has been beneficial in monitoring glucose levels and managing T2D, there are some limitations associated with the technology, including the need for frequent calibration, the high cost of the device, and the discomfort associated with wearing the sensor under the skin. On the contrary, fully non-invasive methods are still being developed as alternatives (for an overview of the latest developments in non-invasive CGM-alternatives, see Laha et al. (2022) [8]). In this light, the use of non-invasive devices or sensing technologies to provide valuable information and vital data monitoring presents a less intrusive and effective option for managing T2D. Therefore, it is necessary to consider whether wearable devices are currently being used effectively in the management and treatment of patients diagnosed with T2D and whether there are studies available that provide evidence of the efficacy of vital data monitoring using these devices.

Non-invasive wearable devices in monitoring vital signs, improving patients’ outcomes or providing valuable insights into an individual’s health have increasingly gained significant attention in recent years. Wearable monitors are electronic devices worn on the body and include patches, clothing-based monitors, chest straps, upper arm bands, and wristbands. The use of these devices have been reviewed in clinical settings [9], clinical trials [10] and outpatient settings [11].

When used in a specific medical context, factors such as patient history, medical condition, accuracy, reliability, and the context being used, need to be carefully considered. A recent review by Prieto-Avalos et al. (2022) has highlighted the considerable number of wearable devices that are increasingly being used to monitor cardiovascular diseases, which can be a complication of T2D. The authors note that heart rate (HR), blood oxygen saturation (SpO_2_), and electrocardiogram technique (ECG) are the biomedical variables most commonly measured in CVD monitoring using commercial wearable devices. However, the review also reveals that not all wearable devices meet the required standards for accuracy and reliability [12].

Other reviews have focused on blood glucose monitoring using smart devices [13,14,15]. A meta-analysis on the use of wearable devices in the treatment of chronic diseases showed benefits for patients with diabetes mellitus or heart disease. However, the focus was on weight reduction, blood glucose, hemoglobin and exercise time [16]. However, the effects of using wearable devices for vital sign monitoring have not been systematically reviewed.

This study aims to fill this research gap by systematically reviewing the literature for studies using telehealth applications or smart devices (certified as medical devices) in RCTs to remotely monitor patient vital signs. Furthermore, most commercially available wearable sensors currently only track physical activities and heart rate. In this review, we consider blood pressure, heart rate, temperature and respiratory rate as vital signs.

We explored the existing applications of wearable devices in managing T2D and elucidated how these devices contribute to the effectiveness of various T2D therapies. Although several studies have investigated the use of wearable devices for monitoring vital signs in T2D [9,11,17], there is still a lack of robust and reliable evidence of their actual use and effects in patients with T2D. The majority of these studies are considered to be of lower quality (i.e., no RCTs), limiting their clinical implications due to inherent risks and uncertainties in their design and procedures. Thus, many limitations still exist in understanding the impact of wearable devices for T2D management, which are partially due to the limited diligence of previous studies. This work will mitigate these limitations and provide ample evidence on and about the usability, utilization and existing and potential impact of the use of wearables for T2D management.

The remainder of this paper is structured as follows: Section 2 lays out the literature search strategy used to identify suitable studies that met our inclusion criteria and the process of extracting relevant data from these studies. Section 3.1 and Section 3.2 detail the study selection and characteristics, respectively. Section 3.3 appraises the methodological quality of the studies and summarizes the main findings. Section 4 evaluates the strengths and weaknesses of the available evidence and provides an overall interpretation of the results considering other evidence. Finally, Section 5 recommends potential areas for future research.

## 2. Methods

Our review followed the guidelines provided by the Preferred Reporting Items for Systematic Reviews and Meta-Analyses (PRISMA) [18].

### 2.1. Systematic Literature Search, Information Sources and Article Selection

We employed the following search strategy to identify related publications (according to the PICOS specifications) by their titles and abstracts in the electronic databases PubMed (https://pubmed.gov, accessed on 8 November 2022), IEEE Xplore (https://ieeexplore.org, accessed on 8 November 2022), and the Cochrane Central Register of Controlled Trials (CENTRAL, https://cochranelibrary.com, accessed on 8 November 2022), as shown in Box 1.

Box 1Search strategy to identify related publications following the PICOS specifications.
**Search Strategy:**
    (“T2D” OR “Type 2 Diabetes” OR “Type 2 Diabetes Mellitus”) AND  (“Sensors” OR “Sensor Devices” OR “Wearables” OR “Wearable Devices”)

All searches were conducted on 8 November 2022. This search strategy was supplemented by filter settings (RCTs, systematic reviews, meta-analyses).

The Population, Intervention, Comparison, Outcome, and Study (PICOS) framework [19] was used as the framework for the literature search strategy in this study. Studies are included if they satisfy all of the following criteria:(i)Studies focusing on patients with type 2 diabetes mellitus;(ii)Studies that used sensors or wearable devices to measure vital signs, including body temperature (BT), blood pressure (BP), heart rate (HR), or respiratory rate (RR);(iii)Studies with any control group;(iv)Studies yielding any outcome;(v)All evidence from randomized controlled trials (RCTs), systematic reviews (SRs), or meta-analyses (MAs) published since the beginning of the year 2017.

Following this, five reviewers (A.P., L.J., J.I.D.O., S.U., and T.N.) independently screened all titles and abstracts. The full text of the studies was further examined if at least one reviewer deemed a publication relevant for the review. Conflicting views were resolved through discussions between the five reviewers. Our search was also extended to the reference lists of all eligible articles. The full text of these selected studies was then analyzed. Two reviewers, assigned in alternating partnerships, checked the relevance of each study for our review according to the inclusion criteria mentioned above. Unanimous decisions were reached in cases of disagreement. All studies that met the criteria were included for data extraction.

### 2.2. Data Extraction, Risk of Bias Assessment Tool and Quality Scales

Two independent reviewers extracted data using an electronic spreadsheet. The extracted data fell under the following categories:Brief description of the study design,Technical details of the device used,Demographic information of the participants andMain results—primary and secondary endpoints.

Assessing bias in studies is crucial as it can explain variations in the results of studies included in a systematic review. We assessed the risk of bias using Cochrane’s recommended tool for randomized controlled trials (RCTs). This tool uses a domain-based evaluation, where appraisals are made separately for different domains. Based on these appraisals, ratings are then assigned to indicate the risk of bias in each domain [20,21]. This was carried out by assigning ‘low risk’, ‘high risk’, or ‘unclear risk’ to each of the domains, namely, selection bias, performance bias, detection bias, attrition bias, and reporting bias. In case of disagreements during the extraction process, the two reviewers (AP and JIDO) discussed until a consensus was reached. If necessary, a third reviewer was consulted. The Cochrane tool was instrumental in our assessment of the risk of bias of RCTs in our systematic review. Regular quality checks ensured consistency between reviewers. Our methods were designed to maintain the integrity of the review process and provide a comprehensive and unbiased review of the wearable devices used to monitor vital signs in patients with type 2 diabetes.

## 3. Results

### 3.1. Study Selection

Figure 1 presents the PRISMA four-phase flow diagrams that illustrate the different stages involved in identifying and selecting studies that focused on applying and reporting sensing technologies for measuring vital signs in patients with type 2 diabetes. The flow diagram is essential to ensure the evaluation is conducted rigorously and precisely. Using the aforementioned search strategies, 437 abstracts were retrieved from three databases—PubMed, IEEE Xplore, and CENTRAL, on the 8 November 2022. After a thorough manual search for duplicate records, 57 instances were identified and removed. Then, an initial screening based on the titles and abstracts was done. 324 records were excluded. The reference lists of the remaining papers were scanned; however, they did not reveal any additional publications for inclusion in the review.

After conducting a comprehensive analysis of the entire text of the research papers, two were study protocols for RCTs with no published results. Despite multiple attempts to contact the authors and garner responses from the studies, their results were unobtainable [22,23]. After careful analysis of the remaining studies, we excluded fortynine due to insufficient evidence, lack of reported vital signs, use of invasive methods, or failure to consider T2D, as illustrated in Figure 1. As a result, only seven studies were finally included in this systematic review.

### 3.2. Study Characteristics

Importantly, two systematic reviews, Price et al. (2022) [24] and Mattison et al. (2022) [25], did not feature any other RCTs relevant for our research, except the study by Frias et al. (2017) [26] which has already been included in our review. As a result, the reviews by Price et al. (2022) and Mattison et al. (2022) were not considered further. Table 1 and Table 2 provide an overview and summary of all studies included in this systematic review of non-invasive wearables for monitoring vital signs in patients with type 2 diabetes mellitus.

#### 3.2.1. Wearable Technology

The studies reviewed employed a variety of sensing technologies and devices to measure physiological vital signs and other health parameters, from ingestible sensors in pills to wearables and smartphones. Heart rate was a universal vital sign consistently monitored across the studies reviewed. Frias et al. (2017) used an ingestible sensor enclosed in a placebo pill to track medication adherence in patients as the pill traveled through the digestive system. The authors also used an adhesive wearable sensor patch and a smartphone to collect data on gastrointestinal activity, body movement and orientation, heart rate, and step count. These data were transmitted to a mobile phone via a dedicated app and a web portal for easy access and analysis [26]. Similarly, Li et al. (2021) employed a wireless chest-worn heart rate monitoring device along with the Recovery Plus Health app. A Lunar iDXA dual-energy X-ray absorptiometer (GE Healthcare, Chicago, IL, USA) was used to assess bone mineral density and body composition, while a hydraulic grip handheld dynamometer (model 12-0240 by Fabrication Enterprises, NY, USA ) measured muscle strength. Participants in the intervention group were equipped with these sensors to accurately measure exercise frequency, intensity, duration, volume, and progression. In their study, the key aim was to determine whether participants reached their target heart rate during physical activities. Cardiorespiratory endurance was also assessed, requiring participants to perform a 3-min YMCA step test that measures an individual’s aerobic capacity [27]. In addition, in studies like the prospective and multicentre RCT by Korn et al. (2021), participants were tasked to perform daily endurance sessions monitored by a heart rate sensor (model H7 heart rate sensor by Polar, Kempele, Finland). The data was recorded in the LeIKD app (IDS Diagnostic Systems, Frankfurt am Main, Germany) [28]. As highlighted in the systematic review by Rodriguez-León et al. (2021), heart rate sensors were employed in 27% of the studies in predicting exercise intensity, calorie consumption, and activity recognition.

Besides vital signs monitoring, various sensing technologies and devices were considered in the reviewed studies including wearables, smartphones, and 30 different types of sensors. Among the reviewed studies, the accelerometer sensor was used most frequently (73%), followed by glucose monitoring sensors (46%). Wearable devices played a crucial role in collecting data related to physical activity, heart rate, blood glucose levels, and other vital health parameters. The monitoring of participants’ activity and health is facilitated through the use of commercial devices like chest straps, lower limb bands, wristbands, and flash glucose monitors. Additionally, three innovative research prototypes, namely a hip-worn strap, a chest strap, and a smart insole, were used to gather physical activity data [29]. The trial conducted by Coombes et al. (2021) broadened the scope by employing a variety of devices. A wristband heart rate monitor (model Lynk2 by Accuro, Oakbrook Terrace, IL, USA) and the Personal Activity Intelligence Health App (by PAI Health, Vancouver, BC, Canada) were used to measure the heart rate and store data, respectively. Various devices were used to monitor physiological variables, including a CGM system, a metabolic analyzer, an electrocardiogram (ECG), a blood pressure monitor, and a body composition analyzer. A bioimpedance scale and photoplethysmography (PPG) sensor were also used. The placement or positioning of these devices on the participant’s body was strategic. The CGM device was worn on the abdomen, ECG electrodes were placed on the chest, the blood pressure monitor was positioned on the upper arm, the portable metabolic analyzer was connected to a face mask or mouthpiece, and the wristband heart rate monitor and PPG sensor were worn on the wrist, during measurements [30].

#### 3.2.2. Outcomes

The primary outcomes of the seven studies included in this systematic review varied depending on the research question, intervention, and technology used. Frias et al. (2017) focused on using digital medicine offerings (DMO) to treat uncontrolled hypertension and type 2 diabetes. The primary outcome was a significant reduction in systolic blood pressure at week 4 and a nonsignificant difference in HbA1c at week 12, with no significant differences in fasting plasma glucose change compared to routine care [26].

Li et al. (2021) conducted a prospective, multicenter RCT to investigate the effects of a mobile app-based exercise program on physical activity levels. The primary outcomes were body fat percentage and cardiorespiratory endurance and the secondary outcomes included blood glucose level, insulin level, homeostasis model assessment of insulin resistance (HOMA-IR), muscle strength, and cholesterol level [27].

In another RCT, von Korn et al. (2021) compared the effects of individualized telemedical-supported lifestyle intervention with routine care over a duration of six months, followed by another six months of follow-up without feedback. The primary outcome observed within groups was the change in HbA1C, while the secondary outcomes measured included health literacy, physical activity, eating behavior, quality of life, cardiovascular risk factors, major cardiovascular events, as well as healthcare costs at the 6 and 12-month marks [28].

The systematic review by Rodriguez-León et al. (2021) assessed the use of mobile and wearable technology in monitoring parameters related to diabetes mellitus, with an emphasis on the prevalence of wearable devices providing continuous measurements. The accelerometer, glucose, and heart rate sensors were the most commonly used sensors. Other analyzed outcomes are attention to privacy and security issues, emerging sensor technologies, and validated clinical trials [29].

Finally, Coombes et al. (2021) evaluated the effectiveness of a lifestyle intervention program in improving glycemic control and various health outcomes in an RCT. The study further assessed the feasibility, acceptability, and efficacy of an e-health software for self-monitoring patients with T2D [30].

### 3.3. Risk of Bias Assessment and Quality Appraisal

The risk of bias summary presented in Figure 2 follows the revised Cochrane risk-of-bias tool for randomized trials (RoB-2) [20]. Our two authors (A.P. and J.I.D.O.) assessed all RoB-2 signaling questions, providing either of the following evaluation options for each question: “yes”, “probably yes”, “probably no”, “no”, or “no information”. For each reviewed trial, the assessment for whether, based on the domain, is categorized as low risk, high risk, or has some concerns is decided. The domains cover the randomization process bias, intervention assignment bias, missing outcome data bias, outcome measurement bias, and reported result selection bias. Based on these domains, an overall risk of bias was determined.

Notably, three RCTs [27,28,30] demonstrated a low risk of bias in the randomization process. While there were concerns for intervention allocation bias across all included studies, the risk of bias due to missing outcome data was low. The outcome measurement bias was high in three studies and medium in one. In terms of selective outcome reporting, one study indicated a low risk of bias, two had a medium risk, and one exhibited a high risk. Figure 2 provides an overview of risk-of-bias assessment for the five domains according to the RoB-2 tool for each of the included RCTs.

As shown in Figure 3, the overall risk of bias is high in three studies and medium in one, hence indicating significant variation in the quality of trial conduct and reporting across the studies analyzed. A more detailed illustration is provided in Appendix A.

## 4. Discussion

This systematic review examined the use of non-invasive wearable devices for monitoring vital signs in patients with T2D. The results indicate that while these devices have considerable potential in monitoring vital signs and managing T2D, there is a need for more robust first-class evidence to validate their practical use and effects in patients with T2D.

Our results are consistent with previous research that has shown the potential of telehealth applications to support the prevention and treatment of chronic diseases, including T2D. However, a closer examination of the studies within this review reveals a reliance on heart rate monitoring devices for primary outcome measurements such as changes in blood pressure, heart rate, HbA1c, or body fat percentage. To address this limitation, we recommend that future studies expand their focus on using more wearable sensors to measure a broader range of vital signs, including body temperature, blood pressure, and respiratory rate. To ensure the meaningfulness of data comparison, it is crucial to establish standardized criteria for comparing data obtained through various sensing technologies.

Some potential criticisms of this review include the limited number of included studies and the risk of bias revealed in several assessment domains, such as outcome measurement bias and bias in the selection of reported results. These biases have most likely influenced the results reported in the studies, making it compelling to interpret the findings cautiously. It has become necessary to address these limitations by conducting high-quality RCTs that rigorously assess the efficacy, safety, and long-term impact of wearable devices in T2D management. These RCTs should also consider addressing methodological limitations, such as randomization process bias, outcome measurement bias, and reporting bias, to enhance the validity and reliability of the findings.

Wearable technology could have economic implications for the healthcare system. For instance, the continuous stream of data provided by these devices could lead to the early detection of complications, potentially reducing hospitalizations and subsequent healthcare costs. For instance, by analyzing patterns in vital sign data, machine learning algorithms may assist in the early detection of fluctuations in glucose levels.

Interestingly, the integration of machine learning in analyzing the massive data streams generated by these wearable devices holds vast potential for T2D management and treatment. Therefore, the integration of machine learning could significantly enhance the applications of these wearable devices by making them more precise, proactive, and patient-centred. Machine learning algorithms, which have previously proven successful in leveraging and analyzing the vast amounts of data collected from wearable sensors to provide personalized insights, potentially play a revolutionary role as predictive models for predicting T2D [32,33,34,35]. Machine learning can help optimize treatment plans, predict individual responses to lifestyle changes, and provide real-time feedback to patients, thus enabling better disease control and personalized and timely interventions in revolutionizing T2D, which can extend to other chronic diseases. Although none of the included studies delivered first-class evidence on the use of machine learning in this context, further research is recommended to validate and optimize predictive models for T2D management.

Wearable devices could empower patients to take a more active role in their disease management, leading to improved self-care and better health outcomes. Therefore, the integration of wearable technology in T2D management is very promising for revolutionizing patient care, and ongoing advancements in this area can substantially improve the lives of patients with T2D.

Finally, despite the immense potential of diabetes health technology using non-invasive monitoring devices for T2D, there is poor evidence of standardized regulations and guidelines to promote the interoperability and data security of mobile health technology. This evidence is also lacking in the reviewed studies because the different studies employed a range of devices to monitor the same vital sign data. Improving interoperability and standardization among wearable devices can ease seamless data sharing and integration. There is a need for regulatory bodies and healthcare organizations to develop guidelines and standards for mobile health apps to ensure safety and clinical validity and ensure the protection of patient information through appropriate data security measures. The development and use of non-invasive methods for monitoring T2D have been constrained by the indirect nature of measurements, the need for calibration, and the challenges of achieving accuracy and usability. There is a crucial need to address these challenges through standardization, improving calibration procedures, enhancing accuracy, and ensuring the suitability of these devices for general use and research in order to unlock their full potential in revolutionizing diabetes care [36].

## 5. Conclusions

This systematic review provides a rigorous evaluation of the use and efficacy of non-invasive wearable devices to monitor vital signs in patients with T2D. We scrutinized the reporting and quality of published research, considering the profound potential implications for patients, healthcare practitioners, and researchers. As effective self-management of T2D demands the continuous need for blood glucose monitoring and adopting lifestyle changes, despite still being in their nascent stage of development, fully non-invasive CGM methods have emerged as a promising approach for managing T2D, mirroring the considerable potential of advancements in telehealth applications.

In our review, we have adhered to quality assurance guidelines and conducted comprehensive examinations of available high-standard studies published in recent years. Nevertheless, this review further highlights the persistent gaps in the literature and the need for additional rigorous and unbiased controlled trials to provide more definitive evidence and facilitate meaningful cross-comparisons among studies.

Furthermore, the limited diversity in sensing technologies and the inherent biases in the current studies pose significant complexities in the accurate interpretation and conclusion of the findings. Future research should prioritize transparency and accuracy in bias reporting to understand the true impact and benefits of using wearable devices in T2D management. With the right application and rigorous research, these technological innovations could revolutionize diabetes care, shifting from periodic clinical check-ups to continuous at-home monitoring, facilitating proactive disease management, and promoting better patient autonomy.

## Figures and Tables

**Figure 1 bioengineering-10-01321-f001:**
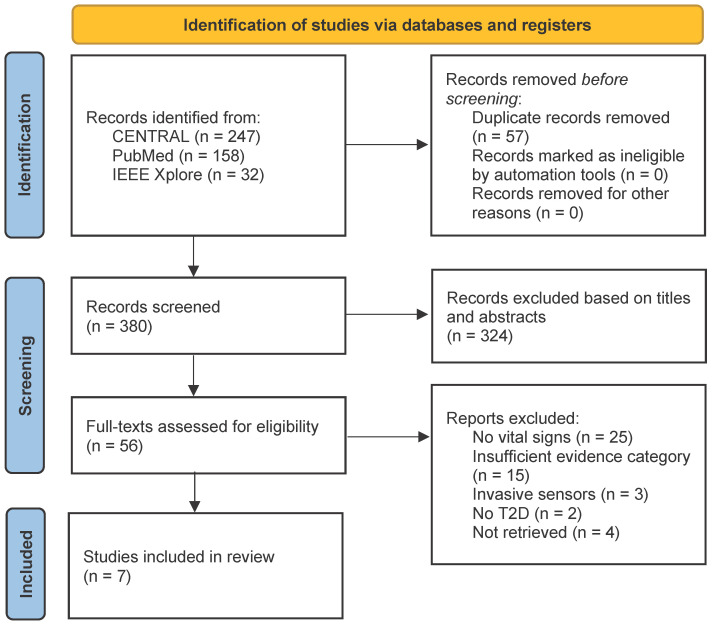
PRISMA four-phase flow diagram outlining identification and selection procedures for the studies included in the qualitative synthesis.

**Figure 2 bioengineering-10-01321-f002:**
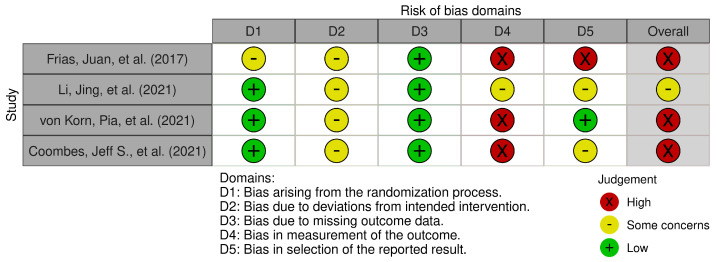
Evaluating trial quality and reporting on the studies by Frias et al. (2017) [26], Li et al. (2021) [27], von Korn et al. (2021) [28], and Coombes et al. (2021) [30]. Risk-of-bias assessment using RoB 2 tool [31] across five domains in included RCTs.

**Figure 3 bioengineering-10-01321-f003:**
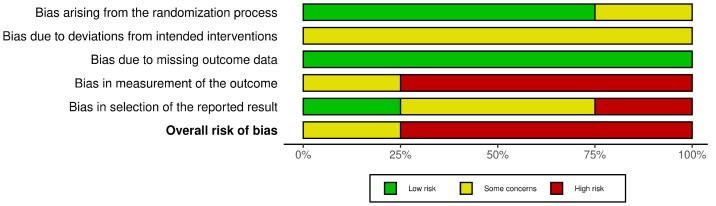
Overview of risk-of-bias assessment using RoB 2 tool [31] across five domains in included RCTs [26,27,28,30] given as percentages.

**Table 1 bioengineering-10-01321-t001:** Overview and summary of all included publications on non-invasive wearables for monitoring vital signs in patients with type 2 diabetes mellitus.

Study	Study Experiment (Aim, Duration, Intervention)	Study Design	Study Population	Vital Signs Measured	Sensing Technology and Devices	Application of the Sensor	Primary Outcome	Secondary Outcomes
Frias, Juan, et al. (2017)	Aim: investigate use of digital medicine in treating T2D Duration: 12 weeks Intervention: combined DMO after failed treatment	RCT	109 participants 13 recruitment sites No information on age	Heart rate	Ingestible sensor inside a placebo pill Adhesive wearable sensor patch Mobile device Web portal	Sensor in pill collects data on medication adherence Wearable patch measures body activity, angle, HR, and step count	Digital med significantly reduced systolic BP at week 4 compared to usual care	Changes in other clinical measures Medication adherence Step count, physical activity and rest duration DMO ns diff in HbA1c redn compared to usual care at week 12 ns diff in fasting plasma glucose chg observed
Li, Jing, et al. (2021)	Aim: effects of mobile app-based exercise program on physical activity levels Intervention: App to track exercise progress	RCT	85 participants 18–64 years	Heart rate	Dynamometer (Fab Ent, model 12-0240) for muscle strength Lunar iDXA dual-energy X-ray absorptiometer (GE Healthcare) for bone mineral density body composition HR band Recovery Plus Inc and Recovery Plus Health app	Measures exercise frequency, intensity, time, volume and progression Determines if participant reaches target heart rate during exercise	Body fat percentage Cardiorespiratory endurance	Blood glucose level Insulin level Homeostasis model assessment of insulin resistance (HOMA-IR) Muscle strength Cholesterol level Intervention group: significantly younger, less likely to have history of hypertension
von Korn, Pia, et al. (2021)	Aim: compare telemedical intervention with usual care Duration: 6 months intervention and 6 months follow-up Intervention: exercise program and counseling, follow-up without counseling	RCT	Different studies with n = 296 to n = 15,487 participants 18+ years	Heart rate	H7 heart rate sensor, Polar, Kempele, Finland	Data recording: LeIKD app	Change in HbA1C (%) after 6 months between groups	Health literacy, physical activity, eating behavior and quality of life Cardiovascular risk factors and major cardiovascular events Healthcare costs at 6 and 12 months

**Table 2 bioengineering-10-01321-t002:** Overview and summary of all included publications on non-invasive wearables for monitoring vital signs in patients with type 2 diabetes mellitus.

Study	Study Experiment (Aim, Duration, Intervention)	Study Design	Study Population	Vital Signs Measured	Sensing Technology and Devices	Application of the Sensor	Primary Outcome	Secondary Outcomes
Rodriguez-León, Ciro, et al. (2021)	Aim: use of mobile and wearable technology for monitoring parameters related to diabetes mellitus	Systematic Review	No information about number of participants and age available	Heart rate	Smartphones Wearables: 30 types of sensors Accelerometer used in 73% of studies Glucose monitoring in 46% of studies Heart rate monitors in 27% of studies	Collect data on: Physical activity Heart rate Blood glucose levels Other Commercial devices: Chest straps Lower limb bands Wristbands Flash glucose monitors Research prototypes: Hip-worn strap Chest strap Smart insole	Mobile and wearable tech for monitoring diabetes mellitus parameters Predominant use of wearables for objective continuous measurements Most commonly used sensors: accelerometer, glucose and heart rate monitors	Wearable devices for diabetes-related monitoring Need for privacy/security attention, emerging sensor tech, and validated trials Few studies on diabetes complications
Coombes, Jeff S., et al. (2021)	Aim: examine feasibility, acceptability, and efficacy of the PAI e-Health program in people with T2D Duration: 12 weeks Intervention: PAI e-Health program	RCT	No information about number of participants 20+ years	Heart rate and blood pressure	CGM, ECG, PPG Metabolic analyzer Blood pressure monitor Wristband heart rate monitor (Lynk2) PAI health app (PAI Health, Vancouver, Canada) Bioimpedance scale	CGM: abdomen Portable metabolic analyzer: face mask/mouthpiece ECG electrodes: chest Blood pressure monitor: upper arm Wristband heart rate monitor: wrist PPG sensor: wrist	Feasibility, acceptability, and efficacy of the PAI e-Health Program	Glycemic control Cardiorespiratory fitness Exercise capacity (time-on-test) Body composition Sleep time Health-related quality of life

## Data Availability

Not applicable.

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
