# Peer review of "Non-Invasive Wearable Devices for Monitoring Vital Signs in Patients with Type 2 Diabetes Mellitus: A Systematic Review"

_bioengineering, 2023, doi:10.3390/bioengineering10111321_

Round 1
Reviewer 1 Report
Comments and Suggestions for Authors
The Authors propose a systematic review on non-invasive wearable devices for monitoring vital signs in patients with type 2 diabetes mellitus. The review has been carried out according to PRISMA e PICOS guidelines. The manuscript has been well written. Here, my comments:
1. In the Introduction, the Authors have to stress more on the novelty of the proposed study.
2. Please clarify the selection of the 7 papers.
3. The review has been carried out in November 2022. Is it possible to update the review?
Author Response
We are deeply grateful to all the reviewers for their invaluable input and high-quality assessment of our manuscript, which has significantly enriched the quality and rigor of our systematic review. We have made diligent efforts to address all the concerns raised by the reviewers to the best of our ability, with the aim of further improving the clarity, validity, and overall strength of our work.
Response to comment 1:
Thank you very much for this comment. To the best of our knowledge, our systematic review is the first to address this specific issue. We have made the following change in the manuscript to make this clear, but were unable to highlight it in the abstract:
Line 6: “This systematic review is the first that explores [...]”
Response to comment 2:
Thank you very much for this comment. We have now moved the selection of the seven publications to the appropriate place in the methodology in the manuscript and listed the inclusion criteria. Accordingly, publications were excluded if they met the following criteria:
- Studies focussing on other diseases
- Studies that do not investigate sensor-based measurement of vital parameters
- Studies without control groups
- Publications that do not present their own results
- Publications that are methodologically not RCTs, SRs or MAs
- Publications in languages other than English or German
- Publications that were not available
The results already included a meaningful text and the PRISMA chart (Fig.1). We have corrected a calculation error in line 147.
Additionally, we have made the following change in the manuscript, but were unable to highlight it in the abstract:
- Line 11: “The seven studies included in this review used various sensing technologies, such as [...]”
Response to comment 3:
Thank you for your comment. Obviously, updating the review is a very tempting thought. However, carrying out the entire systematics of this review again would extend the observation period, but evaluating the data and writing down the results obtained, as well as the associated new review process, would lead to a new gap in time between the survey period and the publication date. However, after receiving your comment, we conducted another search on PubMed and found no new publications relevant to this manuscript. Using our search strategy described in section 2.1, only three results were returned in the period from November 2022 to today (Nov 10, 2023):
- “Within-Person and Between-Sensor Variability in Continuous Glucose Monitoring Metrics”
- “Effects of E-health-based interventions on glycemic control for patients with type 2 diabetes: a Bayesian network meta-analysis”
- “Benefits and Harms of Digital Health Interventions Promoting Physical Activity in People With Chronic Conditions: Systematic Review and Meta-Analysis”
Of these, all three publications do not meet our inclusion criteria described in Section 2.1 and are therefore not relevant for our systematic review. For this reason, this manuscript and all our conclusions are up to date to the best of our knowledge.
Reviewer 2 Report
Comments and Suggestions for Authors
Current manuscript entitled “Non-Invasive Wearable Devices for Monitoring Vital Signs in Patients with Type 2 Diabetes Mellitus: A Systematic Review” by “Piet et al” explores the current literature and critically evaluates the use and reporting of non-invasive wearable devices for monitoring vital signs in patients with Type 2 diabetes mellitus. The studies included in this review used various sensing technologies, such as heart rate monitors, accelerometers, and other wearable devices. Primary health outcomes included blood pressure measurements, heart rate, body fat percentage, and cardiorespiratory endurance. Non-invasive wearable devices demonstrated potential for aiding Type 2 Diabetes Mellitus management, albeit with variations in efficacy across studies. Manuscript seems good and interesting, however substantial changes are required before its acceptance.
1. Authors stated that “Nonetheless, further evidence is required to validate the application, efficacy, and real-world impact of these wearable devices.” How the authors validate this, please state in few lines.
2. Clear statements of novelty should appear briefly in the abstract and conclusions section.
3. In the introduction authors should discuss related review articles published.
4. Captions are missing for the tables. Please cross check.
5. Currently POC and on-site sensing strategies are emerging. Authors should disc these; A simultaneous qualitative and quantitative lateral flow immunoassay for on-site and rapid detection of streptomycin in pig blood serum and urine. Emerging trends in the development of electrochemical devices for the on-site detection of food contaminants. Sowing kernels for food safety: Importance of rapid on‐site detction of pesticide residues in agricultural foods.
6. Provide the challenges that are currently facing with the Non-Invasive Wearable Devices for Type 2 Diabetes Mellitus.
7. The structure of this paper needs to be improved.
8. The manuscript has grammatical errors and typos. Please thoroughly cross check the manuscript.
Comments on the Quality of English LanguageMinor editing of English language required
Author Response
We are deeply grateful to all the reviewers for their invaluable input and high-quality assessment of our manuscript, which has significantly enriched the quality and rigor of our systematic review. We have made diligent efforts to address all the concerns raised by the reviewers to the best of our ability, with the aim of further improving the clarity, validity, and overall strength of our work.
Response to comment 1:
Thank you very much for this comment. To address this, we have made the following change in the manuscript, but were unable to highlight it in the abstract:
- Line 16: “Based on the low number of studies with higher evidence levels (i.e., RCTs) that we were able to find and the significant differences in design between these studies, we conclude that further evidence is required to validate the application, efficacy, and real-world impact of these wearable devices.”
Response to comment 2:
Thank you very much for this comment. To the best of our knowledge, our systematic review is the first to address this specific issue. We have made the following change in the manuscript to make this clear, but were unable to highlight it in the abstract:
- Line 6: “This systematic review is the first that explores [...]”
Response to comment 3:
Thank you for this comment. Following your advice, we have expanded the section in which we had already reported on other reviews.
Response to comment 4:
Thank you for your suggestion. We have added the captions to all tables accordingly.
Response to comment 5:
We thank you for this comment. However, we do not see the direct connection to the objective that we address with this review: a critical assessment of the use and reporting of non-invasive wearable devices for monitoring vital signs in patients with T2D. We do not question the fact that there are other very promising developments in addition to this, however these were not the subject of our analysis.
Response to comment 6:
Thank you for raising this point. We agree with your view and modified the discussion section in our manuscript, emphasizing current challenges of non-invasive wearables for type 2 diabetes mellitus.
Response to comment 7:
We thank you for this comment, which was also made by Reviewer 3. As a consequence, we have reformatted the manuscript, but the basic structure is still based on the PRISMA scheme established for the preparation of systematic reviews.
Response to comment 8:
Thank you for this comment. We have checked the manuscript and corrected all grammatical errors and typos.
Reviewer 3 Report
Comments and Suggestions for Authors The impression given by the review is that the wearable device research field has significant deficiencies. From a total of 437 publications only references 24-28 were judged to have randomized controlled trial credentials. The results should be of interest to researchers, journal editors and funding agencies. The comments below may or may not be of sufficient interest to merit a sentence or two in the text. 1) Like everything else in today's world, there has been inflation in the number of vital signs, from the original 4 to the 5 mentioned on line 87 and perhaps 7 where blood glucose level is included. It's a little unclear in the manuscript where CGM should be positioned. It is not included in the table's vital signs' column but in the technology and applications columns. 2) It is considerably easier to evaluate a review consisting of only five publications, compared to those trying to overwhelm with several hundred. Breaking up the long paragraphs in the text would enhance the forthright delivery of the message. The first paragraph in the introduction, for example, has fifty lines. 3) References in the selected publications were surveyed, but not articles that cited them after they were published. This might be of interest just to see if RCT publications inspire RCT studies. 4) It is outside the scope of the review to analyze why RCT is not common in the wearable device field. Cost would seem to be an obvious reason, but only references 24 and 25 had any commercial funding for their studies. This would seem to indicate that funding agencies are willing to foot the bill for controlled trials and the question arises as to why they don't make them a requirement in order to improve the chance that studies "provide more definite evidence" as stated on line 312. The opposite side of the issue is if the excluded publications had any common underfunded characteristic, such as by an individual hospital or regional funding agency. Or no external funding at all, which is not uncommon in some non-traditional medical fields such as ozone treatment of lower back pain. 5) Motivation would seem to be a key factor in T2D management. It is thus puzzling that access to information generated by wearables seemed to differ between intervention groups. In reference 25, for example, participants were notified during exercise if they were reaching their heart rate target. In 26 they were informed by phone afterwards about their progress. 6) The question is if wearables will ever be able to undergo the same rigourous, double-blind studies that drugs are subjected to, or if this would even be desirable. All subjects could be fitted with devices, but how the information would be withheld from the control group would be a problem. Reference 21 was excluded because of non-reported results. Two groups consisting of early or late night eaters were studied to determine the effect of walking 8000 steps five times per week, making the definition of control somewhat vague. 7) Referring to point 1, it is CGM which has the lead in the field, to the degree that the "promising approach" designation on line 312 may be an understatement. After receiving CE marking in 2014 for its FreeStyle Libre monitoring system, Abbott UK financed multicenter studies which contributed to FDA approval. Dexcom and Medtronic have since entered the field and it appears that academic groups in the area which lack the necessary economic resources to conduct RCT are doomed to conduct studies which will only be able to conclude the need for more studies. However, such non-RCT studies do seem to serve a function as training for the particular application area, and as a basis for which groups commercial interests will choose for their clinical trials when the technology has matured. Most of the academic authors of the Freestyle publications reported previously receiving grants, funding and fees from Abbott Diabetes Care outside the submitted work. Intervention participants in the Abbott studies used wearable, flash glucose monitoring for management, including insulin decisions, while the control group relied on traditional measurement. There thus doesn't seem to be any way to blind such a study with respect to the device employed.Author Response
We are deeply grateful to all the reviewers for their invaluable input and high-quality assessment of our manuscript, which has significantly enriched the quality and rigor of our systematic review. We have made diligent efforts to address all the concerns raised by the reviewers to the best of our ability, with the aim of further improving the clarity, validity, and overall strength of our work.
Response to comment 1:
Dear reviewer, thank you for your comment. In our understanding, vital signs are: temperature, blood pressure, heart rate, and respiratory rate. We focus precisely on these parameters in this systematic review because there is evidence suggesting that supplementing the already well-established CGM with vital signs monitoring can lead to improved disease management. However, we thank you for your comment, which prompted us to formulate this aspect more clearly in the manuscript.
Response to comment 2:
Thank you for this suggestion, which was also made by Reviewer 2. We have reformatted the entire manuscript accordingly.
Response to comment 3:
Thank you very much for this exciting indication of an interesting extension of the research question and the very good methodological impulse. We have not analysed this in this paper, but will take the important information with us for a potential follow-up study.
Response to comment 4:
We agree with your considerations. You are certainly right that it is an interesting question how “commercial funding” influences the planning, design and implementation of corresponding studies. However, we did not have any information for our review, particularly on the question of why there was no funding. Therefore, we were unable to make any valid statements regarding these concerns.
Response to comment 5:
Thank you for your comment. Here, again, we agree with your impression. Nevertheless, based on our review, we are not in a position to make such a statement clearly. Our review aims to investigate which devices are used for vital sign monitoring in RCTs and which results are reported in this context. Of course, you are right that the different studies use different designs, but in our opinion the evidence base - as we compiled it in our review - is not sufficient to draw a conclusion regarding the effect of individual design aspects (e.g., communication of critical values) or to derive possible implications for future study designs or forms of therapy.
Response to comment 6:
Thank you for this important insight. We are very interested in a further exchange outside of the review process. We can certainly envision conducting our own investigation into this matter.
Response to comment 7:
We agree with you about the meaning of CGM you described. However, since there is initial evidence that suggests that a combination of CGM and vital sign monitoring can be a promising addition to the management of diabetes, we asked ourselves whether such vital sign monitoring is currently possible successfully with portable, wearable devices and have therefore carried out this systematic review.
Round 2
Reviewer 2 Report
Comments and Suggestions for Authors
The revised manuscript can be accepted for publication
Comments on the Quality of English LanguageMinor editing of English language required